# Median Selection Subset Aggregation for Parallel Inference

**Xiangyu Wang**
Dept. of Statistical Science
Duke University
xw56@stat.duke.edu

**Peichao Peng**
Statistics Department
University of Pennsylvania
ppeichao@yahoo.com

**David B. Dunson**
Dept. of Statistical Science
Duke University
dunson@stat.duke.edu

## Abstract

For massive data sets, efficient computation commonly relies on distributed algorithms that store and process subsets of the data on different machines, minimizing communication costs. Our focus is on regression and classification problems involving many features. A variety of distributed algorithms have been proposed in this context, but challenges arise in defining an algorithm with low communication, theoretical guarantees and excellent practical performance in general settings. We propose a MEdian Selection Subset AGgregation Estimator (message) algorithm, which attempts to solve these problems. The algorithm applies feature selection in parallel for each subset using Lasso or another method, calculates the 'median' feature inclusion index, estimates coefficients for the selected features in parallel for each subset, and then averages these estimates. The algorithm is simple, involves very minimal communication, scales efficiently in both sample and feature size, and has theoretical guarantees. In particular, we show model selection consistency and coefficient estimation efficiency. Extensive experiments show excellent performance in variable selection, estimation, prediction, and computation time relative to usual competitors.

## 1 Introduction

The explosion in both size and velocity of data has brought new challenges to the design of statistical algorithms. Parallel inference is a promising approach for solving large scale problems. The typical procedure for parallelization partitions the full data into multiple subsets, stores subsets on different machines, and then processes subsets simultaneously. Processing on subsets in parallel can lead to two types of computational gains. The first reduces time for calculations within each iteration of optimization or sampling algorithms via faster operations; for example, in conducting linear algebra involved in calculating likelihoods or gradients [1–7]. Although such approaches can lead to substantial reductions in computational bottlenecks for big data, the amount of gain is limited by the need to communicate across computers at each iteration. It is well known that communication costs are a major factor driving the efficiency of distributed algorithms, so that it is of critical importance to limit communication. This motivates the second type of approach, which conducts computations completely independently on the different subsets, and then combines the results to obtain the final output. This limits communication to the final combining step, and may lead to simpler and much faster algorithms. However, a major issue is how to design algorithms that are close to communication free, which can preserve or even improve the statistical accuracy relative to (much slower) algorithms applied to the entire data set simultaneously. We focus on addressing this challenge in this article.

There is a recent flurry of research in both Bayesian and frequentist settings focusing on the second approach [8–14]. Particularly relevant to our approach is the literature on methods for combining point estimators obtained in parallel for different subsets [8, 9, 13]. Mann et al. [9] suggest using

averaging for combining subset estimators, and Zhang et al. [8] prove that such estimators will achieve the same error rate as the ones obtained from the full set if the number of subsets $m$ is well chosen. Minsker [13] utilizes the geometric median to combine the estimators, showing robustness and sharp concentration inequalities. These methods function well in certain scenarios, but might not be broadly useful. In practice, inference for regression and classification typically contains two important components: One is variable or feature selection and the other is parameter estimation. Current combining methods are not designed to produce good results for both tasks.

To obtain a simple and computationally efficient parallel algorithm for feature selection and co-efficient estimation, we propose a new combining method, referred to as *message*. The detailed algorithm will be fully described in the next section. There are related methods, which were proposed with the very different goal of combining results from different imputed data sets in missing data contexts [15]. However, these methods are primarily motivated for imputation aggregation, do not improve computational time, and lack theoretical guarantees. Another related approach is the bootstrap Lasso (Bolasso) [16], which runs Lasso independently for multiple bootstrap samples, and then intersects the results to obtain the final model. Asymptotic properties are provided under fixed number of features ($p$ fixed) and the computational burden is not improved over applying Lasso to the full data set. Our *message* algorithm has strong justification in leading to excellent convergence properties in both feature selection and prediction, while being simple to implement and computationally highly efficient.

The article is organized as follows. In section 2, we describe *message* in detail. In section 3, we provide theoretical justifications and show that *message* can produce better results than full data inferences under certain scenarios. Section 4 evaluates the performance of *message* via extensive numerical experiments. Section 5 contains a discussion of possible generalizations of the new method to broader families of models and online learning. All proofs are provided in the supplementary materials.

## 2    Parallelized framework

Consider the linear model which has $n$ observations and $p$ features,

$$Y = X\beta + \epsilon,$$

where $Y$ is an $n \times 1$ response vector, $X$ is an $n \times p$ matrix of features and $\epsilon$ is the observation error, which is assumed to have mean zero and variance $\sigma^2$. The fundamental idea for communication efficient parallel inference is to partition the data set into $m$ subsets, each of which contains a small portion of the data $n/m$. Separate analysis on each subset will then be carried out and the result will be aggregated to produce the final output.

As mentioned in the previous section, regression problems usually consist of two stages: feature selection and parameter estimation. For linear models, there is a rich literature on feature selection and we only consider two approaches. The risk inflation criterion (RIC), or more generally, the generalized information criterion (GIC) is an $l_0$-based feature selection technique for high dimensional data [17–20]. GIC attempts to solve the following optimization problem,

$$\hat{M}_\lambda = \arg \min_{M \subset \{1,2,\cdots,p\}} \|Y - X_M \beta_M\|_2^2 + \lambda |M| \sigma^2 \tag{1}$$

for some well chosen $\lambda$. For $\lambda = 2(\log p + \log \log p)$ it corresponds to RIC [18], for $\lambda = (2 \log p + \log n)$ it corresponds to extended BIC [19] and $\lambda = \log n$ reduces to the usual BIC. Konishi and Kitagawa [18] prove the consistency of GIC for high dimensional data under some regularity conditions.

Lasso [21] is an $l_1$ based feature selection technique, which solves the following problem

$$\hat{\beta} = \arg \min_\beta \frac{1}{n} \|Y - X\beta\|_2^2 + \lambda \|\beta\|_1 \tag{2}$$

for some well chosen $\lambda$. Lasso transfers the original NP hard $l_0$-based optimization to a problem that can be solved in polynomial time. Zhao and Yu [22] prove the selection consistency of Lasso under the Irrepresentable condition. Based on the model selected by either GIC or Lasso, we could then apply the ordinary least square (OLS) estimator to find the coefficients.

As briefly discussed in the introduction, averaging and median aggregation approaches possess different advantages but also suffer from certain drawbacks. To carefully adapt these features to regression and classification, we propose the median selection subset aggregation (*message*) algorithm, which is motivated as follows.

Averaging of sparse regression models leads to an inflated number of features having non-zero coefficients, and hence is not appropriate for model aggregation when feature selection is of interest. When conducting Bayesian variable selection, the median probability model has been recommended as selecting the single model that produces the best approximation to model-averaged predictions under some simplifying assumptions [23]. The median probability model includes those features having inclusion probabilities greater than 1/2. We can apply this notion to subset-based inference by including features that are included in a majority of the subset-specific analyses, leading to selecting the 'median model'. Let $\gamma^{(i)} = (\gamma_1^{(i)}, \cdots, \gamma_p^{(i)})$ denote a vector of feature inclusion indicators for the $i^{th}$ subset, with $\gamma_j^{(i)} = 1$ if feature $j$ is included so that the coefficient $\beta_j$ on this feature is non-zero, with $\gamma_j^{(i)} = 0$ otherwise. The inclusion indicator vector for the median model $M_\gamma$ can be obtained by

$$\gamma = \arg \min_{\gamma \in \{0,1\}^p} \sum_{i=1}^{m} \|\gamma - \gamma^{(i)}\|_1,$$

or equivalently,

$$\gamma_j = median\{\gamma_j^{(i)}, i = 1, 2, \cdots, m\} \text{ for } j = 1, 2, \cdots, p.$$

If we apply Lasso or GIC to the full data set, in the presence of heavy-tailed observation errors, the estimated feature inclusion indicator vector will converge to the true inclusion vector at a polynomial rate. It is shown in the next section that the convergence rate of the inclusion vector for the median model can be improved to be exponential, leading to substantial gains in not only computational time but also feature selection performance. The intuition for this gain is that in the heavy-tailed case, a proportion of the subsets will contain outliers having a sizable influence on feature selection. By taking the median, we obtain a central model that is not so influenced by these outliers, and hence can concentrate more rapidly. As large data sets typically contain outliers and data contamination, this is a substantial practical advantage in terms of performance even putting aside the computational gain. After feature selection, we obtain estimates of the coefficients for each selected feature by averaging the coefficient estimates from each subset, following the spirit of [8]. The *message* algorithm (described in Algorithm 1) only requires each machine to pass the feature indicators to a central computer, which (essentially instantaneously) calculates the median model, passes back the corresponding indicator vector to the individual computers, which then pass back coefficient estimates for averaging. The communication costs are negligible.

## 3 Theory

In this section, we provide theoretical justification for the *message* algorithm in the linear model case. The theory is easily generalized to a much wider range of models and estimation techniques, as will be discussed in the last section.

Throughout the paper we will assume $X = (x_1, \cdots, x_p)$ is an $n \times p$ feature matrix, $s = |S|$ is the number of non-zero coefficients and $\lambda(A)$ is the eigenvalue for matrix $A$. Before we proceed to the theorems, we enumerate several conditions that are required for establishing the theory. We assume there exist constants $V_1, V_2 > 0$ such that

A.1 **Consistency condition for estimation.**
- $\frac{1}{n}x_i^T x_i \leq V_1$ for $i = 1, 2, \cdots, p$
- $\lambda_{min}(\frac{1}{n}X_S^T X_S) \geq V_2$

A.2 **Conditions on $\epsilon$, $|S|$ and $\beta$**
- $E(\epsilon^{2k}) < \infty$ for some $k > 0$
- $s = |S| \leq c_1 n^\iota$ for some $0 \leq \iota < 1$

---
**Algorithm 1** *Message* algorithm
---
**Initialization:**
 1: Input $(Y, X), n, p$, and $m$
 2:     *# n is the sample size, p is the number of features and m is the number of subsets*
 3: Randomly partition $(Y, X)$ into m subsets $(Y^{(i)}, X^{(i)})$ and distribute them on m machines.
**Iteration:**
 4: **for** $i = 1$ to $m$ **do**
 5:     $\gamma^{(i)} = \min_{M_\gamma} loss(Y^{(i)}, X^{(i)})$   *# $\gamma^{(i)}$ is the estimated model via Lasso or GIC*
 6: *# Gather all subset models $\gamma^{(i)}$ to obtain the median model $M_\gamma$*
 7: **for** $j = 1$ to $p$ **do**
 8:     $\gamma_j = median\{\gamma_j^{(i)}, i = 1, 2, \cdots, m\}$
 9: *# Redistribute the estimated model $M_\gamma$ to all subsets*
10: **for** $i = 1$ to $m$ **do**
11:     $\beta^{(i)} = (X_\gamma^{(i)T} X_\gamma^{(i)})^{-1} X_\gamma^{(i)T} Y_\gamma^{(i)}$   *# Estimate the coefficients*
12: *# Gather all subset estimations $\beta^{(i)}$*
13: $\bar{\beta} = \sum_{i=1}^{m} \beta^{(i)}/m$
14:
15: **return** $\bar{\beta}, \gamma$
---

- $\min_{i \in S} |\beta_i| \geq c_2 n^{-\frac{1-\tau}{2}}$ for some $0 < \tau \leq 1$

A.3 (Lasso) **The strong irrepresentable condition.**

- Assuming $X_S$ and $X_{S^c}$ are the features having non-zero and zero coefficients, respectively, there exists some positive constant vector $\eta$ such that

$$|X_{S^c}^T X_S (X_S^T X_S)^{-1} sign(\beta_S)| < 1 - \eta$$

A.4 (Generalized information criterion, GIC) **The sparse Riesz condition.**

- There exist constants $\kappa \geq 0$ and $c > 0$ such that $\rho > cn^{-\kappa}$, where

$$\rho = \inf_{|\pi| \leq |S|} \lambda_{min}(X_\pi^T X_\pi/n)$$

A.1 is the usual consistency condition for regression. A.2 restricts the behaviors of the three key terms and is crucial for model selection. These are both usual assumptions. See [19, 20, 22]. A.3 and A.4 are specific conditions for model selection consistency for Lasso/GIC. As noted in [22], A.3 is almost sufficient and necessary for sign consistency. A.4 could be relaxed slightly as shown in [19], but for simplicity we rely on this version. To ameliorate possible concerns on how realistic these conditions are, we provide further justifications via Theorem 3 and 4 in the supplementary material.

**Theorem 1.** *(GIC) Assume each subset satisfies A.1, A.2 and A.4, and $p \leq n^\alpha$ for some $\alpha < k(\tau - \eta)$, where $\eta = \max\{\iota/k, 2\kappa\}$. If $\iota < \tau$, $2\kappa < \tau$ and $\lambda$ in (1) are chosen so that $\lambda = c_0/\sigma^2 (n/m)^{\tau-\kappa}$ for some $c_0 < cc_2/2$, then there exists some constant $C_0$ such that for $n \geq (2C_0 p)^{(k\tau-k\eta)^{-1}}$ and $m = \lfloor (4C_0)^{-(k\tau-k\eta)^{-1}} \cdot n/p^{(k\tau-k\eta)^{-1}} \rfloor$, the selected model $M_\gamma$ follows,*

$$P(M_\gamma = M_S) \geq 1 - \exp\left\{-\frac{n^{1-\alpha/(k\tau-k\eta)}}{24(4C_0)^{(k\tau-k\eta)}}\right\},$$

*and defining $C_0' = \min_i \lambda_{min}(X_\gamma^{(i)T} X_\gamma^{(i)}/n_i)$, the mean square error follows,*

$$E\|\bar{\beta} - \beta\|_2^2 \leq \frac{\sigma^2 V_2^{-1} s}{n} + \exp\left\{-\frac{n^{1-\alpha/(k\tau-k\eta)}}{24(4C_0)^{(k\tau-k\eta)}}\right\}\left((1 + 2C_0'^{-1}sV_1)\|\beta\|_2^2 + C_0'^{-1}\sigma^2\right).$$

**Theorem 2.** *(Lasso) Assume each subset satisfies A.1, A.2 and A.3, and $p \leq n^\alpha$ for some $\alpha < k(\tau - \iota)$. If $\iota < \tau$ and $\lambda$ in (2) are chosen so that $\lambda = c_0(n/m)^{\frac{\iota-\tau+1}{2}}$ for some $c_0 < c_1 V_2/c_2$, then there exists some constant $C_0$ such that for $n \geq (2C_0 p)^{(k\tau-k\iota)^{-1}}$ and $m = \lfloor (4C_0)^{(k\tau-k\iota)^{-1}} \cdot n/p^{(k\tau-k\iota)^{-1}} \rfloor$, the selected model $M_\gamma$ follows*

$$P(M_\gamma = M_S) \geq 1 - \exp\left\{-\frac{n^{1-\alpha/(k\tau-k\iota)}}{24(4C_0)^{(k\tau-k\iota)}}\right\},$$

*and with the same $C_0'$ defined in Theorem 1, we have*

$$E\|\bar{\beta} - \beta\|_2^2 \leq \frac{\sigma^2 V_2^{-1} s}{n} + \exp\left\{ -\frac{n^{1-\alpha/(k\tau - k\iota)}}{24(4C_0)^{(k\tau - k\iota)}} \right\} \left( (1 + 2C_0'^{-1} s V_1)\|\beta\|_2^2 + C_0'^{-1}\sigma^2 \right).$$

The above two theorems boost the model consistency property from the original polynomial rate [20, 22] to an exponential rate for heavy-tailed errors. In addition, the mean square error, as shown in the above equation, preserves almost the same convergence rate as if the full data is employed and the true model is known. Therefore, we expect a similar or better performance of *message* with a significantly lower computation load. Detailed comparisons are demonstrated in Section 4.

## 4 Experiments

This section assesses the performance of the *message* algorithm via extensive examples, comparing the results to

- Full data inference. (denoted as "full data")
- Subset averaging. Partition and average the estimates obtained on all subsets. (denoted as "averaging")
- Subset median. Partition and take the marginal median of the estimates obtained on all subsets (denoted as "median")
- Bolasso. Run Lasso on multiple bootstrap samples and intersect to select model. Then estimate the coefficients based on the selected model. (denoted as "Bolasso")

The Lasso part of all algorithms will be implemented by the "glmnet" package [24]. (We did not use ADMM [25] for Lasso as its actual performance might suffer from certain drawbacks [6] and is reported to be slower than "glmnet" [26])

### 4.1 Synthetic data sets

We use the linear model and the logistic model for $(p; s) = (1000; 3)$ or $(10,000; 3)$ with different sample size $n$ and different partition number $m$ to evaluate the performance. The feature vector is drawn from a multivariate normal distribution with correlation $\rho = 0$ or $0.5$. Coefficients $\beta$ are chosen as,

$$\beta_i \sim (-1)^{ber(0.4)}(8\log n/\sqrt{n} + |N(0,1)|), i \in S$$

Since GIC is intractable to implement (NP hard), we combine it with Lasso for variable selection: Implement Lasso for a set of different $\lambda$'s and determine the optimal one via GIC. The concrete setup of models are as follows,

Case 1 Linear model with $\epsilon \sim N(0, 2^2)$.

Case 2 Linear model with $\epsilon \sim t(0, df = 3)$.

Case 3 Logistic model.

For $p = 1,000$, we simulate 200 data sets for each case, and vary the sample size from 2000 to 10,000. For each case, the subset size is fixed to 400, so the number of subsets will be changing from 5 to 25. In the experiment, we record the mean square error for $\hat{\beta}$, probability of selecting the true model and computational time, and plot them in Fig 1 - 6. For $p = 10,000$, we simulate 50 data sets for each case, and let the sample size range from 20,000 to 50,000 with subset size fixed to 2000. Results for $p = 10,000$ are provided in supplementary materials.

It is clear that *message* had excellent performance in all of the simulation cases, with low MSE, high probability of selecting the true model, and low computational time. The other subset-based methods we considered had similar computational times and also had computational burdens that effectively did not increase with sample size, while the full data analysis and bootstrap Lasso approach both were substantially slower than the subset methods, with the gap increasing linearly in sample size. In terms of MSE, the averaging and median approaches both had dramatically worse

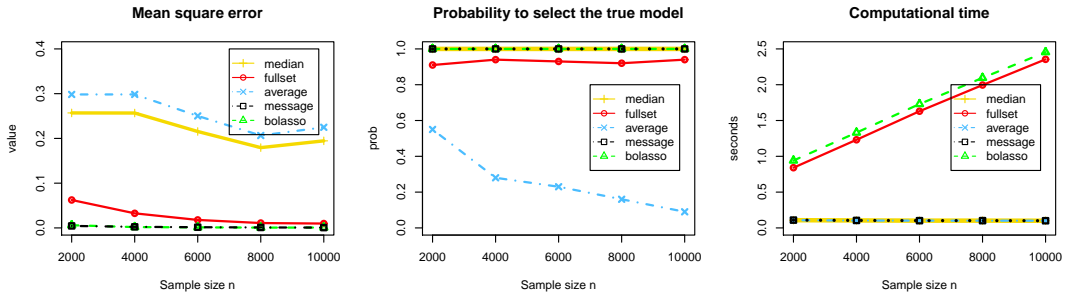

Figure 1: Results for case 1 with $\rho = 0$.

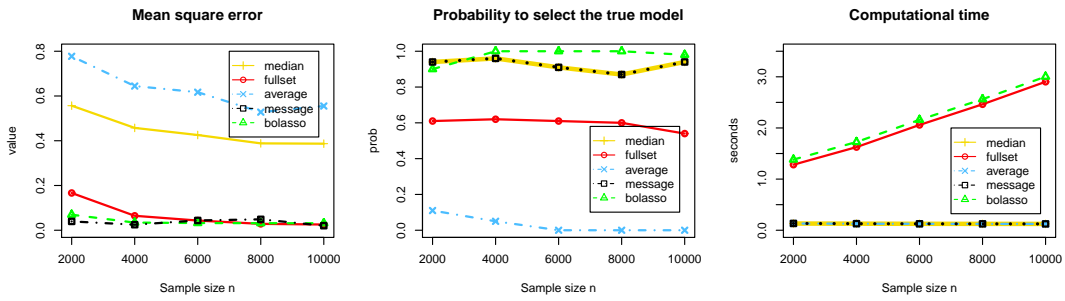

Figure 2: Results for case 1 with $\rho = 0.5$.

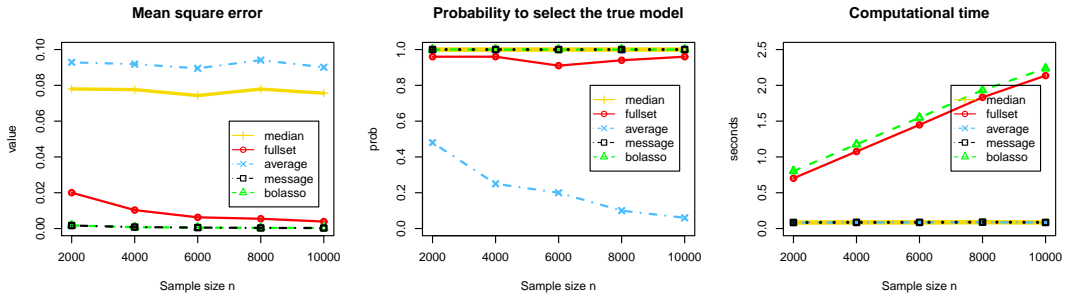

Figure 3: Results for case 2 with $\rho = 0$.

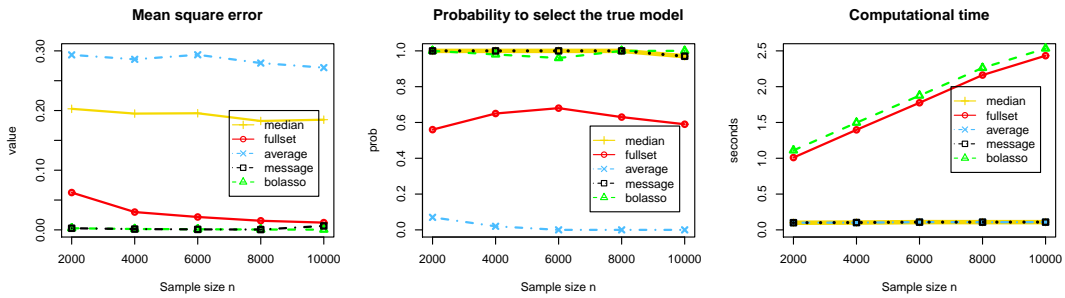

Figure 4: Results for case 2 with $\rho = 0.5$.

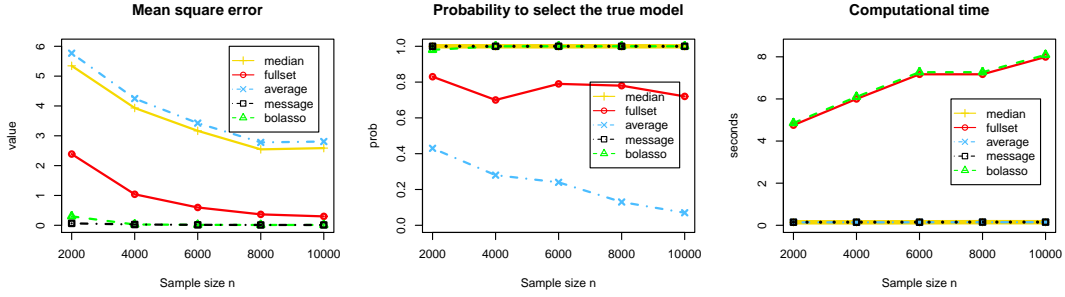

Figure 5: Results for case 3 with $\rho = 0$.

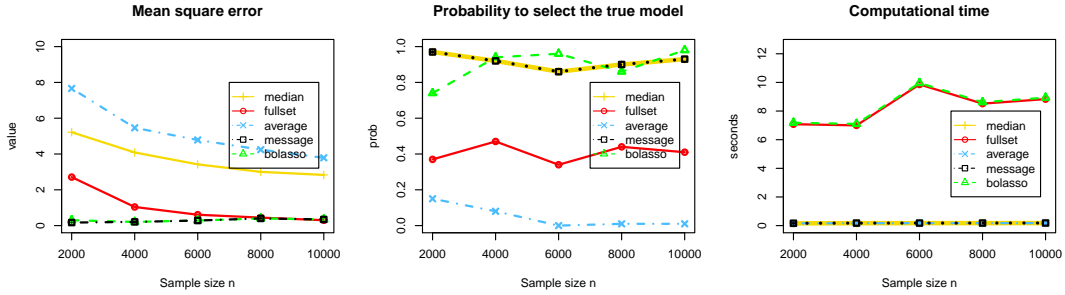

Figure 6: Results for case 3 with $\rho = 0.5$.

performance than *message* in every case, while bootstrap Lasso was competitive (MSEs were same order of magnitude with *message* ranging from effectively identical to having a small but significant advantage), with both *message* and bootstrap Lasso clearly outperforming the full data approach. In terms of feature selection performance, averaging had by far the worst performance, followed by the full data approach, which was substantially worse than bootstrap Lasso, median and *message*, with no clear winner among these three methods. Overall *message* clearly had by far the best combination of low MSE, accurate model selection and fast computation.

### 4.2 Individual household electric power consumption

This data set contains measurements of electric power consumption for every household with a one-minute sampling rate [27]. The data have been collected over a period of almost 4 years and contain 2,075,259 measurements. There are 8 predictors, which are converted to 74 predictors due to re-coding of the categorical variables (date and time). We use the first 2,000,000 samples as the training set and the remaining 75,259 for testing the prediction accuracy. The data are partitioned into 200 subsets for parallel inference. We plot the prediction accuracy (mean square error for test samples) against time for full data, message, averaging and median method in Fig 7. Bolasso is excluded as it did not produce meaningful results within the time span.

To illustrate details of the performance, we split the time line into two parts: the early stage shows how all algorithms adapt to a low prediction error and a later stage captures more subtle performance of faster algorithms (full set inference excluded due to the scale). It can be seen that *message* dominates other algorithms in both speed and accuracy.

### 4.3 HIGGS classification

The HIGGS data have been produced using Monte Carlo simulations from a particle physics model [28]. They contain 27 predictors that are of interest to physicists wanting to distinguish between two classes of particles. The sample size is 11,000,000. We use the first 10,000,000 samples for training a logistic model and the rest to test the classification accuracy. The training set is partitioned into 1,000 subsets for parallel inference. The classification accuracy (probability of correctly predicting the class of test samples) against computational time is plotted in Fig 8 (Bolasso excluded for the same reason as above).

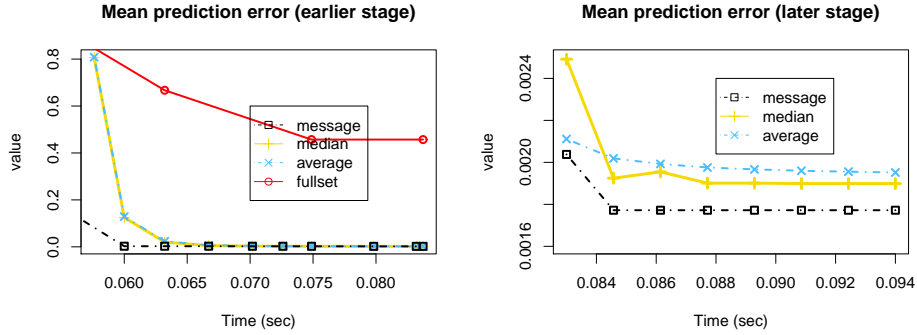

Figure 7: Results for power consumption data.

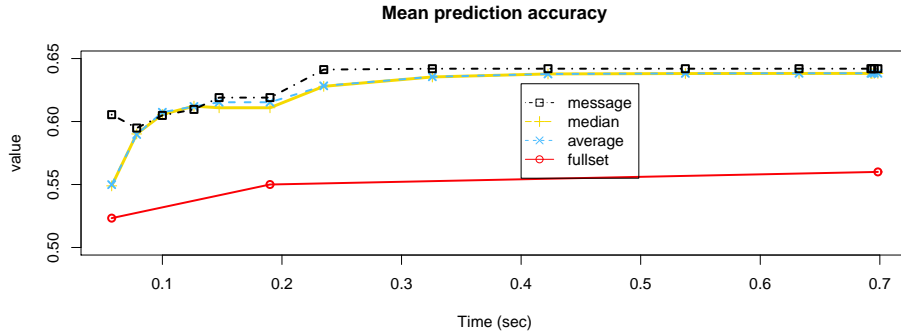

Figure 8: Results for HIGGS classification.

*Message* adapts to the prediction bound quickly. Although the classification results are not as good as the benchmarks listed in [28] (due to the choice of a simple parametric logistic model), our new algorithm achieves the best performance subject to the constraints of the model class.

## 5 Discussion and conclusion

In this paper, we proposed a flexible and efficient *message* algorithm for regression and classification with feature selection. *Message* essentially eliminates the computational burden attributable to communication among machines, and is as efficient as other simple subset aggregation methods. By selecting the median model, *message* can achieve better accuracy even than feature selection on the full data, resulting in an improvement also in MSE performance. Extensive simulation experiments show outstanding performance relative to competitors in terms of computation, feature selection and prediction.

Although the theory described in Section 3 is mainly concerned with linear models, the algorithm is applicable in fairly wide situations. Geometric median is a topological concept, which allows the median model to be obtained in any normed model space. The properties of the median model result from independence of the subsets and weak consistency on each subset. Once these two conditions are satisfied, the property shown in Section 3 can be transferred to essentially any model space. The follow-up averaging step has been proven to be consistent for all M estimators with a proper choice of the partition number [8].

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
