[Supplementary Material · Supplementary_final_dd.pdf]

# Median selection subset aggregation for parallel inference: Supplementary material

**Xiangyu Wang**
Dept. of Statistical Science
Duke University
Durham, NC 27708
xw56@stat.duke.edu

**Peichao Peng**
Statistics Department
University of Pennsylvania
Philadelphia, PA 19104
ppeichao@yahoo.com

**David B. Dunson**
Dept. of Statistical Science
Duke University
Durham, NC 27708
dunson@stat.duke.edu

## Proof of Theorem 1 and Theorem 2

In this section, we prove the following stronger version of Theorem 1 and Theorem 2.

**Theorem 1.** *(GIC) Assume each subset satisfies A.1, A.2 and A.4, and $p \leq n^\alpha$ for some $\alpha < k(\tau - \eta)$, where $\eta = \max\{\iota/k, 2\kappa\}$. If $\iota < \tau$, $2\kappa < \tau$ and $\lambda$ are chosen so that $\lambda = c_0/\sigma^2 (n/m)^{\tau - \kappa}$ for some $c_0 < cc_2/2$, then there exists some constant $C_0$ such that for $n \geq (2C_0 p)^{(k\tau - k\eta)^{-1}}$, any $\delta \in (0, 1/2)$ and $m = \lfloor (\frac{\delta}{C_0})^{(k\tau - k\eta)^{-1}} n/p^{(k\tau - k\eta)^{-1}} \rfloor$, the selected model $M_\gamma$ follows,*

$$P(M_\gamma = M_S) \geq 1 - \exp\left\{ -\frac{(1/2 - \delta)^2}{2(1-\delta)} (\frac{\delta}{C_0})^{(k\tau - k\eta)^{-1}} n^{1 - \alpha(k\tau - k\eta)^{-1}} \right\},$$

*and the mean square error of the aggregated estimator follows,*

$$E\|\bar\beta - \beta\|_2^2 \leq \frac{\sigma^2 V_2^{-1} s}{n} + \exp\left\{ -\frac{(1/2 - \delta)^2}{2(1-\delta)} (\frac{\delta}{C_0})^{(k\tau - k\eta)^{-1}} n^{1 - \alpha(k\tau - k\eta)^{-1}} \right\} \left( (1 + 2C_0'^{-1} s V_1) \|\beta\|_2^2 + C_0'^{-1}\sigma^2 \right).$$

*where $C_0' = \min_i \lambda_{min}(X_\gamma^{(i)T} X_\gamma^{(i)}/n_i)$.*

**Theorem 2.** *(Lasso) Assume each subset satisfies A.1, A.2 and A.3, and $p \leq n^\alpha$ for some $\alpha < k(\tau - \iota)$. If $\iota < \tau$ and $\lambda$ are chosen so that $\lambda = c_0 (n/m)^{\frac{\iota - \tau + 1}{2}}$ for some $c_0 < c_1 V_2/c_2$, then there exists some constant $C_0$ such that for $n \geq (2C_0 p)^{(k\tau - k\iota)^{-1}}$, any $\delta \in (0, 1/2)$ and $m = \lfloor (\frac{\delta}{C_0})^{(k\tau - k\iota)^{-1}} \cdot n/p^{(k\tau - k\iota)^{-1}} \rfloor$, the selected model $M_\gamma$ follows*

$$P(M_\gamma = M_S) \geq 1 - \exp\left\{ -\frac{(1/2 - \delta)^2}{2(1-\delta)} (\frac{\delta}{C_0})^{(k\tau - k\iota)^{-1}} n^{1 - \alpha(k\tau - k\iota)^{-1}} \right\},$$

*and with the same $C_0'$ defined in Theorem 1, we have*

$$E\|\bar\beta - \beta\|_2^2 \leq \frac{\sigma^2 V_2^{-1} s}{n} + \exp\left\{ -\frac{(1/2 - \delta)^2}{2(1-\delta)} (\frac{\delta}{C_0})^{(k\tau - k\iota)^{-1}} n^{1 - \alpha(k\tau - k\iota)^{-1}} \right\} \left( (1 + 2C_0'^{-1} s V_1) \|\beta\|_2^2 + C_0'^{-1}\sigma^2 \right).$$

Fixing $\delta = 1/4$ gives exactly the Theorem 1 and Theorem 2 in the article. The above two theorems can be implied from the following three lemmas.

**Lemma 1.** *(Median model for Lasso) Assume each subset satisfies A.1, A.2 and A.3. If $\iota < \tau$ and $\lambda$ is chosen so that $\lambda = c_0 (n/m)^{\frac{\iota - \tau + 1}{2}}$ for some $c_0 < c_1 V_2/c_2$, then there exists some constant $C_0$ such that for $n \geq (2C_0 p)^{\frac{1}{k(\tau - \iota)}}$ and any $\delta \in (0, 1/2)$, the selected median model $M_\gamma$ satisfies*

$$P(M_\gamma = M_S) \geq 1 - \exp\left\{ -\frac{(1/2 - \delta)^2}{2(1-\delta)} (\frac{\delta}{C_0})^{(k\tau - k\iota)^{-1}} p^{-(k\tau - k\iota)^{-1}} n \right\},$$

*where $\delta$ determines the number of subsets $m$*

$$m = \lfloor (\frac{\delta}{C_0})^{k^{-1}/(\tau-\iota)} n/p^{(k\tau-k\iota)^{-1}} \rfloor$$

*and constant $C_0$ is defined in the proof.*

*In particular, if $p \leq n^\alpha$ for some $\alpha < k(\tau - \iota)$ then*

$$P(M_\gamma = M_S) \geq 1 - \exp\left\{ -\frac{(1/2-\delta)^2}{2(1-\delta)} (\frac{\delta}{C_0})^{(k\tau-k\iota)^{-1}} \cdot n^{1-\alpha(k\tau-k\iota)^{-1}} \right\}.$$

**Lemma 2.** *(Median model for GIC) Assume each subset satisfies A.1, A.2 and A.4. Let $\eta = \max\{\iota/k, 2\kappa\}$. If $\iota < \tau$, $2\kappa < \tau$ and $\lambda$ is chosen so that $\lambda = c_0/\sigma^2 (n/m)^{\tau-\kappa}$ for some $c_0 < cc_2/2$, then there exists some constant $C_0$ such that for any $\delta \in (0, 1/2)$ and $n \geq (2C_0 p)^{(k\tau-k\eta)^{-1}}$, the selected median model $M_\gamma$ satisfies*

$$P(M_\gamma = M_S) \geq 1 - \exp\left\{ -\frac{(1/2-\delta)^2}{2(1-\delta)} (\frac{\delta}{C_0})^{(k\tau-k\eta)^{-1}} \cdot p^{(k\tau-k\eta)^{-1}} n \right\},$$

*where $\delta$ determines the number of subsets $m$*

$$m = \lfloor (\frac{\delta}{C_0})^{(k\tau-k\iota)^{-1}} n/p^{(k\tau-k\iota)^{-1}} \rfloor.$$

*In particular, when $p \leq n^\alpha$ for some $\alpha < k(\tau - \eta)$ we have*

$$P(M_\gamma = M_S) \geq 1 - \exp\left\{ -\frac{(1/2-\delta)^2}{2(1-\delta)} (\frac{\delta}{C_0})^{(k\tau-k\eta)^{-1}} n^{1-\alpha(k\tau-k\eta)^{-1}} \right\}.$$

Recall that the design matrix for the true model $M_S$ is assumed to be positive-definite (Assumption A.1) for all subsets. It is therefore reasonable to ensure the selected model $M_\gamma$ possess the same property, and thus we have,

**Lemma 3.** *(MSE for averaging) Assume $\hat{\beta}_i$ is the OLS estimator obtained from each subset based on the selected model $\gamma$, then the averaged estimator $\bar{\beta}$ has the mean square error,*

$$E\left[ \|\bar{\beta} - \beta\|_2^2 \,|M_\gamma = M_S \right] \leq \frac{\sigma^2 V_2^{-1} s}{n}.$$

*and*

$$E\left[ \|\bar{\beta} - \beta\|_2^2 \,|M_\gamma \neq M_S \right] \leq (1 + 2C_0'^{-1} s V_1) \|\beta\|_2^2 + C_0'^{-1}\sigma^2.$$

*where $C_0' = \min_i \lambda_{min}(X_\gamma^{(i)T} X_\gamma^{(i)}/n_i)$.*

**Proof of Lemma 1**

*Proof.* Following the proof of Theorem 3 in [1], we have the following result: for the $i^{th}$ subset, the selected model $M_{\gamma^{(i)}}$ follows that,

$$P(M_{\gamma^{(i)}} = M_S) \geq 1 - C_1(\frac{2}{c_1})^{2k} n_i^{-k\tau+\iota} - C_2 \frac{4^k p n_i^k}{\lambda^{2k}}, \tag{1}$$

where $C_1, C_2$ are constants:

$$C_1 = \frac{c_1(2k-1)!!E(\epsilon^{2k})}{V_2}, \qquad C_2 = (2k-1)!!V_1 E(\epsilon^{2k})$$

and $n_i = n/m$. If $\lambda$ is chosen to be $c_0(n/m)^{\frac{\tau-\iota+1}{2}}$, then the above result can be updated as

$$P(M_{\gamma^{(i)}} = M_S) \geq 1 - C_1(\frac{2}{c_1})^{2k} n_i^{-k\tau+\iota} - C_2(2/c_0)^{2k} pn_i^{-k\tau+k\iota} \tag{2}$$

$$\geq 1 - C_0 pn_i^{-k\tau+k\iota}, \tag{3}$$

where $C_0$ equals to

$$C_0 = C_1(\frac{2}{c_1})^{2k} + C_2(2/c_0)^{2k}.$$

For any fixed $m$ if the sample size $n$ satisfies

$$n \geq m(2C_0 p)^{\frac{1}{k(\tau-\iota)}},$$

then we have $P(M_{\gamma^{(i)}} = M_S) > 1/2$ on each subset. Recall the definition for the median model,

$$\gamma = \min_{\gamma \in \{0,1\}^p} \sum_{i=1}^m \|\gamma - \gamma^{(i)}\|_1.$$

Notice that as long as half subsets select the correct model, i.e., $card(\{i : M_{\gamma^{(i)}} = M_S\}) \geq m/2$, we will have $M_\gamma = M_S$. Therefore, letting $S_{cor} = \sum_{i=1}^m I_{\{M_{\gamma^{(i)}}=M_S\}}$, where $I_A$ is the indictor function for $A$, we have

$$P(M_\gamma = M_S) = P(S_{cor} \geq \lceil m/2 \rceil). \tag{4}$$

Since all subsets are independent and the correct selection probability for each subset is greater than $1/2$, we can apply the Chernoff inequality ( [2] or Proposition A.6.1 of [3]) to obtain that,

$$P(M_\gamma = M_S) \geq 1 - \exp\left\{ -\frac{(1/2 - C_0 p(n/m)^{-k\tau+k\iota})^2}{2(1 - C_0 p(n/m)^{-k\tau+k\iota})} \cdot m \right\}. \tag{5}$$

Equivalently, for any $n \geq (2C_0 p)^{\frac{1}{k(\tau-\iota)}}$, if we choose $m = \lfloor (\frac{\delta}{C_0})^{k^{-1}/(\tau-\iota)} \cdot n/p^{k^{-1}/(\tau-\iota)} \rfloor$ for any $\delta \in (0, 1/2)$, we have

$$P(M_\gamma = M_S) \geq 1 - \exp\left\{ -\frac{(1/2 - \delta)^2}{2(1 - \delta)} (\frac{\delta}{C_0})^{(k\tau-k\iota)^{-1}} p^{-(k\tau-k\iota)^{-1}} n \right\}.$$

In particular, when $p \leq n^\alpha$ for any $\alpha < k(\tau - \iota)$ we have

$$P(M_\gamma = M_S) \geq 1 - \exp\left\{ -\frac{(1/2 - \delta)^2}{2(1 - \delta)} (\frac{\delta}{C_0})^{(k\tau-k\iota)^{-1}} n^{1-\alpha(k\tau-k\iota)^{-1}} \right\}. \tag{6}$$

$\square$

**Proof of Lemma 2**

*Proof.* The proof is essentially the same as Lemma 1. Following the proof of Theorem 2 in [4], we will have the initial result on each subset,

$$P(M_{\gamma^{(i)}} = M_S) \geq 1 - C_1(\frac{2}{c_2})^{2k} n_i^{-k\tau+\iota} - C_2(\frac{2}{\sigma})^{2k} \frac{p}{(\rho\lambda)^k}, \tag{7}$$

where $n_i = n/m$ and

$$C_1 = \frac{c_1(2k-1)!!E(\epsilon^{2k})}{V_2}, \qquad C_2 = (2k-1)!!V_1 E(\epsilon^{2k}).$$

Now because $\lambda = \frac{c_0}{\sigma^2}(\frac{n}{m})^{\tau-\kappa}$ we update the above equation to

$$P(M_{\gamma^{(i)}} = M_S) \geq 1 - C_1(\frac{2}{c_2})^{2k} n_i^{-k\tau+\iota} - C_2(\frac{4}{c_0 c_3})^k p n_i^{-k(\tau-2\kappa)} \tag{8}$$

$$\geq 1 - C_0 p n_i^{-k(\tau-\eta)}, \tag{9}$$

where $\eta = \max\{\iota/k, 2\kappa\}$ and

$$C_0 = C_1(\frac{2}{c_2})^{2k} + C_2(\frac{4}{c_0 c_3})^k.$$

With exactly the same argument as in Lemma 1, once the sample size exceeds $(2C_0 p)^{\frac{1}{k(\tau-\eta)}}$, then for any $\delta \in (0, 1/2)$ and $m = \lfloor (\frac{\delta}{C_0})^{k^{-1}/(\tau-\eta)} \cdot n/p^{k^{-1}/(\tau-\eta)} \rfloor$, we have

$$P(M_\gamma = M_S) \geq 1 - \exp\left\{ -\frac{(1/2 - \delta)^2}{2(1 - \delta)} (\frac{\delta}{C_0})^{(k\tau-k\eta)^{-1}} p^{-(k\tau-k\eta)^{-1}} n \right\}. \tag{10}$$

In particular, when $p \leq n^\alpha$ for some $\alpha < k(\tau - \eta)$ we have

$$P(M_\gamma = M_S) \geq 1 - \exp\left\{ -\frac{(1/2 - \delta)^2}{2(1 - \delta)} (\frac{\delta}{C_0})^{(k\tau-k\eta)^{-1}} n^{1-\alpha(k\tau-k\eta)^{-1}} \right\}. \tag{11}$$

$\square$

**Proof of Lemma 3**

*Proof.* To simplify the notation, let $X$ denote the selected feature matrix $X\gamma$. Now if the selected model is correct, the error of OLS estimator can be described in the following form,

$$\hat{\beta} - \beta = (X^T X)^{-1} X^T \epsilon. \tag{12}$$

Hence the error of averaged estimator is,

$$\bar{\beta} - \beta = \sum_{i=1}^{m} (X^{(i)T} X^{(i)})^T X^{(i)T} \epsilon^{(i)}/m. \tag{13}$$

Because $E\epsilon^2 = \sigma^2$ we have

$$E\|\bar{\beta} - \beta\|_2^2 = \frac{\sigma^2}{m^2} \sum_{i=1}^{m} tr[(X^{(i)T} X^{(i)})^{-1}]. \tag{14}$$

As the smallest eigenvalue for each subset feature matrix is lower bounded by $V_2$, i.e.,

$$\lambda_{min}(X^{(i)T} X^{(i)}/n_i) \geq V_2,$$

we have

$$E\|\bar{\beta} - \beta\|_2^2 = \frac{\sigma^2}{m^2 n_i} \sum_{i=1}^{m} tr[(X^{(i)T} X^{(i)}/n_i)^{-1}]$$

$$\leq \frac{\sigma^2}{m^2 n_i} \sum_{i=1}^{m} s V_2^{-1}$$

$$= \frac{\sigma^2 V_2^{-1} s}{n}. \tag{15}$$

However, if the model is incorrect, we can bound the incorrect estimators in the following way. For each subset,

$$\|\hat{\beta} - \beta\|_2^2 = \|\hat{\beta}_\gamma - \beta_\gamma\|_2^2 + \|\hat{\beta}_{S/\gamma} - \beta_{S/\gamma}\|_2^2 \leq \|\hat{\beta}_\gamma - \beta_\gamma\|_2^2 + \|\beta_{S/\gamma}\|_2^2. \tag{16}$$

Now to quanitify the first term, we first notice that

$$(X\hat{\beta}_\gamma - X\beta_\gamma)/\sqrt{n_i} = (X(X^T X)^{-1} X^T Y - X\beta_\gamma)/\sqrt{n_i}, \tag{17}$$

and therefore

$$C_0'\|\hat{\beta}_\gamma - \beta_\gamma\|_2^2 \leq \|(X\hat{\beta}_\gamma - X\beta_\gamma)/\sqrt{n_i}\|_2^2 \leq n_i^{-1}(\|Y\|_2^2 - 2\beta_\gamma^T X^T Y + \|X\beta_\gamma\|_2^2) \tag{18}$$

Taking expectation on both sides we have,

$$E\|\hat{\beta}_\gamma - \beta_\gamma\|_2^2 \leq C_0'^{-1}(E\|Y\|_2^2/n_i + \|X_{S/\gamma}\beta_{S/\gamma}\|_2^2)/n_i$$

$$\leq C_0'^{-1}(\|X_S\beta_S\|_2^2/n_i + \sigma^2 + \|X_{S/\gamma}\beta_{S/\gamma}\|_2^2/n_i)$$

$$\leq C_0'^{-1}(2\|\beta\|_2^2 \lambda_{max}(X_S^T X_S/n_i) + \sigma^2)$$

$$\leq C_0'^{-1}(2s V_1 \|\beta\|_2^2 + \sigma^2) \tag{19}$$

Therefore, we have

$$E\|\hat{\beta} - \beta\|_2^2 \leq (1 + 2C_0'^{-1} s V_1)\|\beta\|_2^2 + C_0'^{-1}\sigma^2. \tag{20}$$

The above bound holds for all subset estimtors $\beta^{(i)}$, it should also hold for their average $\bar{\beta}$, i.e.,

$$E\|\bar{\beta} - \beta\|_2^2 \leq (1 + 2C_0'^{-1} s V_1)\|\beta\|_2^2 + C_0'^{-1}\sigma^2. \tag{21}$$

$\square$

## Assumption justification

It is important to carefully assess whether the conditions assumed to obtain theoretical guarantees can be satisfied in applications. There is typically no comprehensive answer to this question, as the answer usually varies across application areas. Nonetheless, we provide some discussion below to provide some insights, while being limited by the complexity of the question.

In following paragraphs, we attempt to justify A.1, A.3 and A.4 with examples and theorems. The main reason to leave A.2 alone is because A.2 is an assumption on basic model structure that is routine in the high-dimensional literature. See Zhao and Yu (2006) and Kim. et al. (2012).

The discussion is divided into two parts. In the first part, we consider the case where features or predictors are independent. In the second part, we will address the correlated case. Because we can always standardize feature matrix X prior to any analysis, it will be convenient to assume $x_{ij}$ having mean 0 and variance 1. For independent features, we have the following result.

**Theorem 3.** *If the entries of the $n \times p$ feature matrix $X$ are i.id random variables with finite $4w^{th}$ moments for some integer $w > 0$, then A.1, A.3 and A.4 will hold for all $m$ subsets with probability,*

$$P(A.1, A.3 \text{ and } A.4 \text{ hold for all subsets}) \geq 1 - O\left\{ \frac{m^{2w}(2s-1)^{2w}p^2}{n^{2w-1}} \right\},$$

*where $s$ is the number of non-zero coefficients.*

*Alternatively, for a given $\delta_0 > 0$, if the sample size $n$ satisfies that*

$$n \geq m(2s-1)\left\{ 9^w(2w-1)!!M_1(2s-1)mp^2\delta_0^{-1} \right\}^{\frac{1}{2w-1}},$$

*where $M_1$ is some constant, then with probability at least $1 - \delta_0$, all subsets satisfy A.1, A.3 and A.4.*

The proof will be provided in next section. Theorem 3 requires $m = o(n)$, which seems to conflict with Theorem 1 and 2 in the article where $m$ is assumed to be $O(n)$ if $p$ is fixed to be constant. This is, however, caused by the choice of $\delta$ (see the stronger version of Theorem 1 provided in the first section). $\delta$ is fixed at $1/4$ in the article for simpicity, leading to the conclusion of $m = O(n)$. With a different choice of $\delta$ satisfying $\delta = o(n)$, Theorem 3 along with Theorem 1 and 2 (stronger version) can be satisfied simultaneously. The same argument can be applied to Theorem 4 introduced in the next part as well.

Next, we consider the case when features are correlated. For data sets with correlated features, preprocessing such as preconditioning might be required to satisfy some of the conditions. Due to the complexity of the problem, we restrict our attention to data sets following elliptical distributions and under high dimensional setting ($p > n$). Real world data commonly follow elliptical distributions approximately, with density proportional to $g(x^T\Sigma^{-1}x)$ for some non-negative function $g(\cdot)$. The Multivariate Gaussian is a special case with $g(z) = exp(-z/2)$. Following the spirit of Jia and Rohe (2012), we make use of $(XX^T/p)^{-1/2}$ ($XX^T$ is invertible when $p > n$) as preconditioning matrix and then use results from Wang and Leng (2014) to show A.1, A.3 and A.4 hold with high probability. Thus, we have the following result.

**Theorem 4.** *Assume $p > n$ and define $\tilde{X} = (XX^T/p)^{-1/2}X$ and $\tilde{Y} = (XX^T/p)^{-1/2}Y$. If each row of feature matrix $X$ are i.i.d samples drawn from an elliptical distribution with covariance $\Sigma$ and the condition number of $\Sigma$ satisfies that $cond(\Sigma) < M_2$ for some $M_2 > 0$, then for any $M > 0$ there exist some $M_3, M_4 > 0$ such that, A.1, A.3 and A.4 hold for all subsets with probability,*

$$P(A.1, A.3 \text{ and } A.4 \text{ hold for all subsets}) \geq 1 - O\left\{ mp^2 \exp\left( \frac{-Mn}{2m\log n} \right) \right\},$$

*if $n \geq \exp\left( 4M_3M_4(2s-1)^2 \right)$.*

*Alternatively, for any $\delta_0 > 0$, if the sample size satisfies that*

$$n \geq \max\left\{ O\left( 2m(\log m + 2\log p - \log \delta_0)/M \right), \ \exp\left( 4M_3M_4(2s-1)^2 \right) \right\},$$

*then with probability at least $1 - \delta_0$, A.1, A.3 and A.4 hold for all subsets.*

The proof is also provided in next section.

**Proof of Theorem 3**

*Proof.* Let $C = \frac{1}{n}X^T X$. We divide the proof into four parts. In Part I and II, we examine the magnitude of $C_{ik}$ and $C_{ii}$ and give the probability that A.3 and the first part of A.1 hold for a single data set. In Part III, we give the probability that A.4 and the second part of A.1 hold on a single data set. We generalize the result to multiple data sets in Part IV.

***Part I***. For $C_{ii}$ we have

$$EC_{ii} = E\|x_i\|_2^2/n = \frac{1}{n}\sum_{j=1}^n Ex_{ij}^2 = 1,$$

and

$$E|C_{ii} - 1|^{2w} = \frac{1}{n^{2w}}E|\sum_{j=1}^n (x_{ij}^2 - 1)|^{2w} \leq \frac{(2w-1)!!E|x_{12}^2 - 1|^{2w}}{n^{2w-1}},$$

where the last inequality follows the proof of Theorem 3 in [1]. Because $E|x_{12}|^{4w} < \infty$, it is clear that $E|x_{12}^2 - 1|^{2w} = \sum_{j=0}^{2w}(-1)^j C_{2w}^j E|x_{12}|^{4w-2j}$ is also a finite value, which will be denoted by $M_0$. Now by the Chebyshev's inequality we have for any $t > 0$,

$$P(|C_{ii} - 1| > t) < \frac{(2w-1)!!M_0}{n^{2w-1}t^{2w}}.$$

Therefore, by taking the union bound over $i = 1, 2, \cdots, p$ we have,

$$P(\max_{i \in \{1,2,\cdots,p\}}|C_{ii} - 1| > t) < \frac{(2w-1)!!M_0 p}{n^{2w-1}t^{2w}}. \tag{22}$$

Recall the definition of $C_{ii}$, (22) immediately implies that the first part of A.1 will hold with probability at least $1 - O\left(\frac{p}{n^{2w-1}}\right)$ for a single data set.

***Part II***. Following the same argument, we can establish the same inequality for $C_{ik}$ where the only difference is the mean,

$$EC_{ik} = Ex_i^T x_k/n = \frac{1}{n}\sum_{j=1}^n Ex_{ij}x_{jk} = 0.$$

From Chebyshev's inequality we have for any $t > 0$,

$$P(|C_{ik}| > t) < \frac{(2w-1)!!M_1}{n^{2w-1}t^{2w}},$$

where $M_1 = \max\{E|x_{12}x_{23}|^{2w}, M_0\}$ is a constant. Taking union bound over all off-diagonal terms we have,

$$P(\max_{i\neq k}|C_{ik}| > t) < \frac{(2w-1)!!M_1 p^2}{n^{2w-1}t^{2w}}. \tag{23}$$

With (22) and (23), we can quantify the sample correlation between $x_i$ and $x_k$, which is $C_{ik}/\sqrt{C_{ii}C_{kk}}$. Taking $t = (6s-3)^{-1}$ for both inequalities, we have

$$P(\max_{i\neq k}|cor(x_i, x_k)| > \frac{1}{4s-2}) < 2(2w-1)!!9^w M_1 \frac{(2s-1)^{2w}p^2}{n^{2w-1}}.$$

With Corollary 2 in [1], the above result essentially states that A.3 will hold with probability at least $1 - O\left(\frac{(2s-1)^{2w}p^2}{n^{2w-1}}\right)$ for a single machine.

***Part III***. For the second part of A.1 and A.4, we might need to quantify the minimum value of $v^T C v$ for any vector $\|v\|_2 = 1$ with support $|supp\{v\}| \leq s$. Here $supp\{a\}$ stands for all non-zero coordinates of vector $a$. Let $S$ be index set for non-zero coefficients. Noticing that,

$$\lambda_{min}(\frac{1}{n}X_S^T X_S) = \min_{\|v\|=1, supp\{v\}=S} v^T C v \geq \min_{\|v\|=1, |supp\{v\}|\leq s} v^T C v,$$

and

$$\inf_{|\pi|\leq s} \lambda_{min}\left(\frac{1}{n}X_\pi^T X_\pi\right) = \inf_{|\pi|\leq s} \min_{\|v\|=1, supp\{v\}=\pi} v^T C v = \min_{\|v\|=1, |supp\{v\}|\leq s} v^T C v.$$

Thus, evaluating $\min_{\|v\|=1, |supp\{v\}|\leq s} v^T C v$ solely is adequate. In fact, for any vector $v$ with $|supp\{v\}| \leq s$ we have,

$$v^T C v = \sum_{i\in supp\{v\}} C_{ii} v_i^2 + \sum_{i\neq k\in supp\{v\}} C_{ik} v_i v_k$$

$$\geq \min_{i\in\{1,2,\cdots,p\}} C_{ii} - \left\{(s^2-s) \max_{i\neq k\in\{1,2,\cdots,p\}} C_{ik}^{(S)2}\right\}^{1/2}. \tag{24}$$

The second step is an application of Cauchy-Schwarz inequality and the fact that if $\|v\|_2 = 1$ then $\sum_{i\neq k} v_i^2 v_k^2 < 1$. Combining (24) with (22) and (23), and taking $t = (2s+2)^{-1}$ we have

$$\min_{\|v\|=1, |supp\{v\}|\leq s} v^T C v \geq 1 - \frac{1}{2s+2} - \frac{s}{2s+2} = \frac{1}{2},$$

with probability at least

$$1 - \frac{2^{2w+1}(2w-1)!!M_1(s+1)^{2w}p^2}{n^{2w-1}}.$$

***Part IV***. Consequently, A.1, A.3 and A.4 will hold for data set on a single machine with probability,

$$P(A.1, A.3 \text{ and } A.4) \geq 1 - O\left\{\frac{(2s-1)^{2w}p^2}{n^{2w-1}}\right\}.$$

Now if we have $m$ subsets, each with sample size $n/m$, then the probability that all subsets satisfy A.1, A.3 and A.4 follows,

$$P(A.1, A.3 \text{ and } A.4 \text{ hold for all}) \geq 1 - O\left\{\frac{m^{2w}(2s-1)^{2w}p^2}{n^{2w-1}}\right\}.$$

Alternatively, for a given $\delta_0 > 0$ and the number of subsets $m$, if the sample size $n$ satisfies that

$$n \geq m(2s-1)\left\{9^w(2w-1)!!M_1(2s-1)mp^2\delta_0^{-1}\right\}^{\frac{1}{2w-1}},$$

then with probability at least $1 - \delta_0$, all subsets satisfy A.1, A.3 and A.4. $\qquad\square$

**Proof of Theorem 4**

*Proof.* The proof procedure is essentially the same as Theorem 3. We begin by looking at data set on a single machine. Let $C = \tilde{X}^T\tilde{X}/n = p/n \cdot X^T(XX^T)^{-1}X$. From Lemma 4 and Lemma 5 of [5] we have, for any $M > 0$ there exists some constant $M_3, M_4 > 0$ such that,

$$P\left(\max_{i\in\{1,2,\cdots,p\}} C_{ii} > M_3 \text{ or } \min_{i\in\{1,2,\cdots,p\}} C_{ii} < M_3^{-1}\right) < 4p \cdot e^{-Mn}, \tag{25}$$

and

$$P\left(\max_{i\neq k\in\{1,2,\cdots,p\}} C_{ik} > \frac{M_4}{\sqrt{\log n}}\right) \leq O\left\{p^2 \cdot \exp\left(\frac{-Mn}{2\log n}\right)\right\}. \tag{26}$$

Inequality (25) implies the first part of A.1. For A.3, we can use (25) and (26) to bound the sample correlation $cor(x_i, x_k) = C_{ik}/\sqrt{C_{ii}C_{kk}}$ as follows,

$$P\left(\max_{i\neq k\in\{1,2,\cdots,p\}} |cor(x_i, x_k)| > \frac{M_3 M_4}{\sqrt{\log n}}\right) \leq O\left\{p^2 \cdot \exp\left(\frac{-Mn}{2\log n}\right)\right\}.$$

Therefore, to satisfy A.3 only requires

$$n \geq \exp\left(4M_3 M_4(2s-1)^2\right). \tag{27}$$

As $s$ is assumed to be small, (27) will not be a big threat to the sample size. For A.4 and the second part of A.1, we continue to apply the same strategy in the proof of Theorem 3 (Part III). Using (24) we have,

$$\min_{\|v\|=1,|supp\{v\}|\leq s} v^T C v \geq M_3^{-1} - \frac{M_4 s}{\sqrt{\log n}}$$

with probability $1 - O\{p^2 \cdot \exp(-Mn/2\log n)\}$. To satisfy A.4 and the second part of A.1, we just need $n$ to be greater than $\exp(M_3 M_4 s^2)$, which is already true if (27) holds.

Consequently, for a single machine if (27) is satisfied, A.1, A.3 and A.4 hold with probability,

$$P(A.1, A.3 \text{ and } A.4) \geq 1 - O\left\{p^2 \cdot \exp\left(\frac{-Mn}{2\log n}\right)\right\}.$$

Now for $m$ subsets, each with sample size $n/m$, the probability that A.1, A.3 and A.4 hold for all subsets follows,

$$P(A.1, A.3 \text{ and } A.4 \text{ hold for all}) \geq 1 - O\left\{mp^2 \cdot \exp\left(\frac{-Mn}{2m\log n}\right)\right\}.$$

Alternatively, for any $\delta_0 > 0$, if

$$n \geq \max\left\{O\left(2m(\log m + 2\log p - \log \delta_0)/M\right), \exp\left(4M_3 M_4 (2s-1)^2\right)\right\},$$

then A.1, A.3 and A.4 hold for all subsets with probability at least $1 - \delta_0$. $\qquad\square$

## Results for $p = 10,000$

we simulate 50 data set for each case, and let the sample size range from 20,000 to 50,000 with subset size fixed to 2,000. Bolasso is not implemented as the computation cost is too expensive. The results are plotted in Fig 1 - 6.

Figure 1: Results for case 1 with $\rho = 0$.

Figure 2: Results for case 1 with $\rho = 0.5$.

Figure 3: Results for case 2 with $\rho = 0$.

Figure 4: Results for case 2 with $\rho = 0.5$.

Figure 5: Results for case 3 with $\rho = 0$.

Figure 6: Results for case 3 with $\rho = 0.5$.