[Reviews · NeurIPS 2014]

Submitted by Assigned_Reviewer_5

The paper has a very simple message. To train distributed linear classifiers or regressors with minimal communication overheads: a) identify the optimal feature set on each machine; b) select the features which are identified on a majority of the machines; c) estimate the weight parameters for each selected feature in parallel on each machine; d) average the estimated weights centrally to get the final classifier.

The main theoretical contribution is a couple of algorithms showing model (feature) selection consistency and parameter estimation efficiency -- under certain conditions such as heavy tail errors. My main concern is whether those conditions are realistic in large scale problem settings eg web link prediction or document classification.

The main allure of the paper is the utterly elegant simplicity of its message.
Summary: Thank you for a very enjoyable paper - it was a pleasure to read such a well written paper!

The main allure of the paper is the utterly elegant simplicity of its message. The authors' response has addressed my earlier concerns from the original review very well.

Submitted by Assigned_Reviewer_35

This paper presents a new method to scale up training of large-scale linear models, where the tacit assumption is that the parameter set fits in a single machine but not all training instances do. The idea starts by splitting the training set into m disjoint subsets and allocating each to a different machine. Then sparsity-inducing regularised risk minimisation (such as lasso or GIC) is run independently in each machine, thus recovering a support set of features in each machine (say s_im=1 if feature i in machine m survived the selection and 0 otherwise). Then a critical step: find a consensus support by picking for each feature the median of its survival indicator value over all machines (ie compute median_m s_im for all features i). Then run OLS with those features independently in each machine and average the resulting models.

This simple, elegant and intuitive approach seems to yield impressive results. The accuracy of both support recovery and prediction are on par with bootstrap lasso, while being much more scalable (constant rather than linear since embarrassingly paralellizable).

I'm also impressed by the quality and clarity of presentation.

The theoretical results presented are precisely those that matter: accuracy of support recovery and accuracy of the estimator.

Empirical results are presented both for suitably designed synthetic experiments and interesting real-world datasets.

Some minor points:

1 - I dind't find a definition of \lambda_{min}. I suppose it's the minimum eigenvalue (?) this should be made clear since notation overloads with \lambda for regularization parameter.
2 - In the algorithm description, I presume you use OLS in line 11 (which is hinted at in the end of page 2). Please make this explicit in the algorithm description (you use "loss" to denote different losses in lines 5 and 11, and this causes confusion)
3 - I think you never give details about *how* the dataset should be split. I feel it needs to be randomly (is that correct?). Could you please make this clear.
4 - Second sentence of introduction starts with lower-case.

Overall a really good paper.
Summary: The paper presents a method to scale up training of linear models. The idea is original, presentation is excellent, empirical results compelling, and all this accompanied with relevant theoretical guarantees.

Submitted by Assigned_Reviewer_45

This paper addresses the model parameter estimation problem when using parallel interference and presents a Median Selection Subset Aggregation Estimator (message) algorithm. The algorithm applies feature selection in parallel for each subset, and calculates the median feature inclusion index, estimates coefficients for the selected features in parallel for each subset, and then averages these estimates. Theoretical analysis shows how the model approximates the true model parameters. Empirical results demonstrate the superiority of the proposed algorithm over both synthetic data and real data.

Overall this paper is well written, and the idea is very clear. The presented approach solves a fundamental problem. It is worth studying if the MESSAGE algorithm can be extended to other models.
Summary: The idea is clear. The method presented in this paper solves a fundamental problem and can attract a lot of attention.
Author Feedback
Author rebuttal: Responses to Reviewer_35

We are very grateful for your constructive comments. Our point-by-point response is as follows.

1. Thanks for pointing out the notation confusion. Yes, \lambda_{min} is the minimum eigenvalue. This will be clarified in the revision.

2. Thanks for the suggestion. Our intention for using "loss'' here is to clarify that the algorithm can be generalized beyond linear regression. However, given that the current version focuses primarily on linear regression, "loss'' is likely to cause certain confusions, so we instead avoid this terminology on line 11 and focus on OLS in the revision.

3. Thanks for pointing out the lack of clarity. Yes, data sets should be partitioned randomly, though it is of future interest to study an optimal method for partitioning even given our excellent results presented in the paper for random partitioning.

Responses to Reviewer_45

We are very grateful for your constructive comments. It is straightforward to extend "message" to much broader settings than linear models, and we conjecture that the algorithm should work well under very general settings. In the paper, we also obtained very good performance for logistic regression.

Responses to Reviewer_5

We are very grateful for your constructive comments. It is important to carefully assess whether the conditions assumed to obtain theoretical guarantees are "realistic" in applications, though it is routine to rely on somewhat overly-restrictive conditions in order to make the theory tractable. Of course, there is typically no comprehensive answer to whether conditions are realistic, as this usually varies across application areas. Nonetheless, we provide some discussion below which should provide some insight, while being limited by the complexity of the question and space restrictions.

The first thing we want to clarify is that the heavy-tailed error is not an assumption in this paper. It is mentioned because "message'' is resistant to outliers. When the error is light-tailed, both “message” and the full data inference work fine for model selection while “message” is much faster. However, "message" will outperform the full data inference in both accuracy and speed when the error is heavy-tailed.

In following paragraphs, we attempt to answer how realistic A.1, A.3 and A.4 are, with examples and justifications. The main reason to leave A.2 alone is because A.2 is an assumption on basic model structure that is routine in the high-dimensional literature. See Zhao and Yu (2006) and Kim. et al. (2012).

The discussion is divided into two parts. In the first part, we consider the case where features or predictors are independent. In the second part, we will address the correlated case. Because we can always standardize feature matrix X prior to any analysis, it will be convenient to assume x_{ij} having mean 0 and variance 1. For independent features, we have the following result.

Theorem 1. If the entries of the n by p feature matrix X are iid random variables with finite 4w-th moment for some integer w > 0, then A.1, A.3 and A.4 will hold for all m subsets with probability,

P(A.1, A.3 and A.4 hold for all) > 1 - O(m^{2w}(2s-1)^{2w}p^2/n^{2w-1}),

where s is the number of non-zero coefficients.
Alternatively, for a given \delta > 0, if the sample size n satisfies that

n > m(2s-1)(9^w(2w-1)!!M_1(2s-1)mp^2\delta^{-1})^{1/(2w-1)},

where M_1 is some constant, then with probability at least 1-\delta, all subsets satisfy A.1, A.3 and A.4.

The proof will be provided in the revised supplementary material.

Next, we consider the case when features are correlated. For data sets with correlated features, pre-processing such as preconditioning might be required to satisfy some of the conditions. Due to the complexity of the problem, we restrict our attention to data sets following elliptical distributions and under high dimensional setting (p > n). Real world data commonly follow elliptical distributions approximately, with density proportional to g(x^T\Sigma^{-1}x) for some non-negative function g(.). The Multivariate Gaussian is a special case with g(z) = exp(-z/2). Following the spirit of Jia and Rohe (2012), we make use of (XX^T/p)^{-1/2} (XX^T is invertible when p > n) as preconditioning matrix and then use results from A. Anonymous (2014) to show A.1, A.3 and A.4 hold with high probability. Thus, we have the following result.

Theorem 2. Assume p > n and define X1 = (XX^T/p)^{-1/2}X and Y1 = (XX^T/p)^{-1/2}Y. If each row of feature matrix X are iid samples drawn from an elliptical distribution with covariance \Sigma and the condition number of \Sigma satisfies that cond(\Sigma) < M_2 for some M_2 > 0. Then for any M > 0 there exist some M_3, M_4 > 0 such that, A.1, A.3 and A.4 hold for X1 in all subsets with probability,

P(A.1, A.3 and A.4 hold for all) > 1 - O(mp^2 exp(-M n/(2m log n))),

if n > exp (4M_3M_4(2s-1)^2).
Alternatively, for any \delta > 0, if the sample size satisfies that

n > max{O(2m(log m + 2log p - log delta)/M), exp(4M_3M_4(2s-1)^2)},

then with probability at least 1-\delta, A.1, A.3 and A.4 hold for all subsets.

The proof will also be provided in the revised supplementary material.